# Weight Loss and Hypertension in Obese Subjects

**DOI:** 10.3390/nu11071667

**Published:** 2019-07-21

**Authors:** Francesco Fantin, Anna Giani, Elena Zoico, Andrea P. Rossi, Gloria Mazzali, Mauro Zamboni

**Affiliations:** Department of Medicine, Section of Geriatrics, University of Verona Healthy Aging Center, Verona, Piazzale Stefani 1, 37126 Verona, Italy

**Keywords:** hypertension, weight loss, obesity

## Abstract

Arterial hypertension is strongly related to overweight and obesity. In obese subjects, several mechanisms may lead to hypertension such as insulin and leptin resistance, perivascular adipose tissue dysfunction, renal impairment, renin-angiotensin-aldosterone-system activation and sympathetic nervous system activity. Weight loss (WL) seems to have positive effects on blood pressure (BP). The aim of this review was to explain the mechanisms linking obesity and hypertension and to evaluate the main studies assessing the effect of WL on BP. We analysed studies published in the last 10 years (13 studies either interventional or observational) showing the effect of WL on BP. Different WL strategies were taken into account—diet and lifestyle modification, pharmacological intervention and bariatric surgery. Although a positive effect of WL could be identified in each study, the main difference seems to be the magnitude and the durability of BP reduction over time. Nevertheless, further follow-up data are needed: there is still a lack of evidence about long term effects of WL on hypertension. Hence, given the significant results obtained in several recent studies, weight management should always be pursued in obese patients with hypertension.

## 1. Introduction

Arterial hypertension is considered one of the most important cardiovascular (CV) risk factors and its connection to overweight and obesity has been extensively proved [1,2]. The prevalence of hypertension among obese patients may range from 60% to 77%, increasing with body–mass index (BMI), in all age groups [3] and it is significantly higher compared to the 34% found in normal weight subjects [3]. These percentages are relevant even when compared to high blood pressure (BP) prevalence in the general population: in 2015, the global age-standardized prevalence was 24.1% (21.4–27.1) in men and 20.1% (17.8–22.5) in women [4]. As shown from the Framingham Heart Study [5], weight gain is responsible for a large percentage of hypertension and it is associated with higher risk of having high BP, even when occurring late in life [6].

The latest definition of hypertension, provided by European Guidelines, is focused on the level of BP (considering office BP, as measured during medical evaluation) at which the benefits of treatment offset its risk, as documented by clinical trials [7]. BP ranges are also defined: the last classification identifies three grades of hypertension (beginning from grade 1, with systolic BP 140–159 mmHg and diastolic BP 90–99 mmHg, followed by grade 2, with SBP 160–179 mmHg and DBP 100–109 mmHg and grade 3, with SBP ≥ 180 mmHg and DBP ≥ 110 mmHg) and isolated systolic hypertension (SBP ≥ 140 and DBP lower than 90 mmHg) [7]. Regardless of the grade, treatment, either with lifestyle interventions or drugs [7], is indicated.

Both obesity and hypertension are considered CV risk factors, therefore their combined management is of utmost importance [8]. It has been shown that moderate weight loss (WL) has a BP lowering effect in both hypertensive and non-hypertensive patients [9]. Furthermore, the magnitude of WL correlates with better results in terms of CV risk reduction [10]. In obese patients with metabolic syndrome, a moderate WL improves renal function [11] and may lead to a 15% reduction of all-cause mortality [12].

Both the latest American and European hypertension guidelines underline the effect of lifestyle modification [7,13] as the first step to be considered in all patients with hypertension and of course in overweight and obese patients. According to the European Society of Cardiology, weight reduction and WL maintenance are mandatory lifestyle changes [7]. American Heart Association highlights that among obese patients, reducing body weight (BW) can lower the risk of developing hypertension to the level of those patients who have never been obese [13]. Moreover, American obesity guidelines report a dose-response effect of the magnitude of WL on BP reduction [14]. Given the strong connection between obesity, overweight and BP and the strong evidence of the possible benefits that can be obtained through weight reduction, the underlying mechanisms and WL strategies should be further explained.

## 2. Mechanisms Linking Obesity to Hypertension

The pathophysiology of hypertension in obese subjects should be considered as a complex phenomenon. The cardiovascular system is affected by structural, functional and hemodynamical changes [15] which can directly increase hypertension risk. The consequences of these changes will not be discussed in the following paragraphs, since they are beyond the purpose of this review. Our attention is focused on the role of different kinds of adipose tissue. The role of Renin Angiotensin Aldosterone System (RAAS) and sympathetic nervous system activation is also considered.

### 2.1. Visceral Adipose Tissue

Fat distribution has been shown to be strongly related to cardiovascular morbidity and mortality [16], independent of the other classical CV risk factors. The distribution of adipose tissue is one of the factors which links obesity to hypertension, along with age of onset of obesity, its duration and degree and weight variation across lifespan [17] (Figure 1). Visceral adiposity, in fact, plays a central role in BP increase, through a greater release of free fatty acid (FFA) in systemic circulation and a consequent increase in insulin resistance and hyperinsulinemia (Figure 1). These changes are firmly related to augmented arterial stiffness and a decrease in vasodilation [17]. Although insulin is a vasodilator hormone, insulin resistance can reduce insulin vasodilation capacity, thereby reducing the nitric oxide (NO) production by endothelial cells [18,19]. Also, the increased levels of insulin are responsible for lumbar SNA promotion, through brain receptor pattern activation, which is directly involved in BP increase [20]. Hyperinsulinemia is found to precede the onset of hypertension in high risk patients and this corroborates the hypothesis of the effect of insulin resistance on BP increase [17].

Furthermore, a strong association has been reported between visceral adipose tissue and greater serum levels of cytokine, such as leptin, interleukin-6, plasminogen activator inhibitor-1, all of which are related both to endothelial dysfunction and hypertension [21,22,23]. The inflammation pattern promoted by cytokines release is involved in an inflammation-dependent aortic stiffening [24] and it can also lead to left ventricular stiffness and mass increase [24]. This hypothesis is well described in the clinical model of metabolic syndrome [24]. Moreover, all the components of metabolic syndrome are shown to be related to augmented carotid-femoral pulse wave velocity [25,26], whereas the same relation cannot be described with the cardio-ankle vascular index [25].

Even if classical effects of leptin include food intake reduction and increasing energy expenditure due to leptin’s central action on the hypothalamus, leptin receptors are also located in the vessels and mainly in the aorta [27], as well as in tunica media and adventitia of arteries and inside atherosclerotic plaques [28]. Through these receptors, leptin may promote vascular smooth muscle cell proliferation and migration, contributing to arterial stiffness [29]. Leptin has been shown also to promote angiogenesis and to activate immune system (both monocytes and T-cell); it is also involved in atherogenesis onset, by increased platelet aggregation and Radical Oxygen Species (ROS) production [30]. As demonstrated in experimental studies in human cells models, leptin also induces endothelial oxidative stress and reactive oxygen species formation [31,32], mechanisms known to increase the risk to develop hypertension. The increased level of leptin is often associated with hypoadiponectinemia [33]: visceral fat, in particular, has also been shown to be negatively associated with adiponectin levels, whose protective effect on arteries is known [34].

### 2.2. Perivascular Adipose Tissue

Perivascular adipose tissue (PVAT) represents adipose tissue (AT) that surrounds blood vessels. Its main function is to provide mechanical support to vessels and even regulate vascular homeostasis. In experimental models of angiotensin II-induced hypertension and deoxycorticosterone acetate (DOCA)-salt hypertension, complement cascade activation is described [35]. In particular, the effector C5a is recognized to promote macrophage infiltration of PVAT, which is responsible for further inflammatory activation [35].

It has been widely shown that PVAT releases biologically active adipokines, with paracrine effects on the vessels [36] (e.g., leptin, adiponectin, omentin, visfatin, resistin, apelin), cytokines/chemokines (e.g., interleukin-6, IL-6; tumour necrosis factor-α, TNF-α; monocyte chemoattractant protein-1, MCP-1), NO, prostacyclin, angiotensin-1 to 7 Angiotensin II and reactive oxygen species (ROS) [37,38,39,40,41]. Obesity leads to a dysfunction of PVAT which releases elevated levels of pro-inflammatory factors adipokines such as leptin, cytokines and chemokines directly to the vascular wall, contributing to endothelial dysfunction and inflammation [39].

All these molecules act with different effects on vascular tone regulation. PVAT-derived relaxing factors (adiponectin, NO, H2S prostacylcin) promote vasodilation and on the other hand PVAT-derived contractile factors, such as leptin, Ang II and ROS induce vasoconstriction. In obese subjects PVAT dysfunction results in augmented production of contractile factors, inducing increased arterial vasoconstriction and greater vascular tone [39,42]. PVAT anticontractile activity is shown to be reduced in hypertensive patients [41].

Moreover, it has been shown that the expression of factors involved in immune cell infiltration and even vascular smooth muscle cells (VSMC) proliferation, are increased in obese subjects, which leads to a general state of inflammation of PVAT and thickening of the arterial wall and probably contributes to an increased risk of hypertension. It has been observed that during the progression of hypertension, immune cells accumulate mainly in perivascular fat tissue surrounding both large and resistance vessels such as the aorta and mesenteric arteries. In particular, one study showed that in non-obesity-induced hypertension inflammation is highly pronounced in PVAT, whereas in non-perivascular visceral fat, immune cell infiltration is less pronounced [42,43,44].

### 2.3. Renal Adipose Tissue

In obese subjects, renal-pressure natriuresis may be impaired through mechanical compression of the kidneys by fat in and around the kidneys [1,45,46]. Increased sodium reabsorption caused by adipose renal tissue mechanical compression could indirectly contribute to renal vasodilation, glomerular hyperfiltration and stimulate increased renin secretion in obese subjects. Moreover, ectopic fat accumulation in and around kidneys seems to have “lipotoxic” effects on kidneys through increased oxidative stress, mitochondrial dysfunction and endoplasmic reticulum stress [47].

Furthermore, natriuresis can be also affected by the activation of the renin-angiotensin-aldosterone system and increased sympathetic nervous system activity, especially by the renal sympathetic nerve activity. (Figure 1). A neural pathway has been hypothesized between renal fat and sympathetic activation [48]. A study by Shi et al. showed an increase in renal sympathetic outflow following enhanced afferent signals from adipose tissue, leading to increased arterial BP in rats. The authors called this reflex the “adipose afferent reflex” (AAR) [49].

Moreover, it has been demonstrated that intra-adipose administration of capsaicin, bradykinin, adenosine or leptin can activate the afferent nerves and consequent AAR [50,51,52], showing a greater enhancement in hypertensive rats compared to normotensive ones. These data seem to show that AAR might be a contributing factor in the pathogenesis of obesity-related hypertension. Altered activities of the adipose-innervating sensory neurons could regulate the cardiovascular system via neural reflex and enhanced hypertension. Nevertheless, more studies are needed to confirm the role of perirenal AAR activation in hypertension pathogenesis, since the anatomical distribution and function of the primary afferent neurons innervating perirenal fat still remains unclear.

### 2.4. Renin Angiotensin Aldosterone System

An important role explaining the increased risk of hypertension in obese patients is surely played by the activation of the renin angiotensin aldosterone system (RAAS) [53,54,55]. In obese subjects, increased renal adipose tissue activates the RAAS through mechanical compression in the kidney. Also, the RAAS can be activated by the increased Sympathetic Nervous System (SNS) activity of obese subjects. (Figure 1). Interestingly, it has been hypothesized that angiotensinogen, produced even by adipocytes, may play a role in determining increased BP in obesity [20,53], even if there is a lack of studies showing a direct effect of angiotensinogen or angiotensin II on BP regulation in obesity. Furthermore, there is evidence that adipocytes can synthesize aldosterone and it may be involved in a paracrine control of vascular function [20].

### 2.5. Sympathetic Nervous System Activation

Several studies showed an increased sympathetic activity in obese subjects, as assessed by direct recordings of muscle sympathetic nerve activity (MSNA) [56,57,58,59,60]. Grassi et al. showed that both heart rate and MSNA baroreflex changes were attenuated in hypertensive obese subjects compared to normotensive subjects. They concluded that the association between obesity and hypertension triggers sympathetic activation together with baroreflex cardiovascular control, which could contribute to the increased incidence of hypertension in obese subjects. Finally, increased levels of leptin, together with the increased levels of pro-inflammatory cytokines, activate the SNS, leading to a BP increase in obese subjects.

Figure 1 describes the complex network which links obesity to arterial hypertension. The effect of different kinds of adipose tissue is shown—PVAT directly promotes local inflammation and contributes to endothelial dysfunction. Visceral AT can induce leptin and insulin resistance, which increase both systemic vasoconstriction and endothelial dysfunction. Moreover, visceral adipose tissue can directly activate the sympathetic nervous system. Perirenal adipose tissue, through mechanical compression, is involved in the RAAS activation and in renal sympathetic system activity. All these pathways, widely interrelated among each other, lead to increased arterial BP in obese subjects.

## 3. Weight Loss and Blood Pressure

Several studies showed that WL may reduce BP: we analysed the studies looking at the effect of WL on BP in the last 10 years; we found 13 studies (either interventional or observational) that showed an association between WL and BP decrease (Table 1). Different WL strategies were taken into account: diet and lifestyle modification, pharmacological intervention and bariatric surgery. Although a positive effect of WL could be identified in each study, the main difference seems to be the magnitude and the durability of BP reduction over time. Some of these results were also corroborated by the evidence of 8 reviews published in the last ten years.

Interestingly, all the studies providing specific WL strategies in obese hypertensive patients showed a significant improvement either in BP decrease and or in body weight reduction. On the contrary, an observational study led by Ho et al., on 2906 obese subjects who developed incident hypertension and achieved BP control within 12 months after diagnosis, showed that the majority of patients did not achieve a significant WL [8]. Therefore, combined management should be pursued.

### 3.1. Diet and Lifestyle Modification

According to published guidelines [7,13], diet and lifestyle modifications aimed at BW reduction are the first step in treating hypertension (COR I, LOE A) [13]. Weight reduction may reduce blood pressure and delay the need of pharmacological antihypertensive therapy [7] and it is also recommended in order to control other associated metabolic risk factors [7]. Straznicky et al. evaluated whether energy restriction could reduce BP in a group of 59 patients affected by obesity. Subjects were treated with dietary intervention or dietary intervention with moderate-intensity aerobic exercise or no treatment, for a period of 12 weeks. In both groups, BW reduction was associated with a significant systolic BP decrease and sympathetic neural activity downregulation [11].

Rothberg et al. enrolled obese patients in a 2-year, intensive, behavioural, weight management program [61] and showed that waist circumference (WC) reduction was related to metabolic syndrome component improvement. After subdividing the study population according to the amount of WC reduction, they observed higher systolic BP reduction in subjects with greater WC decrease, both at the 6-month and 2-year follow-up.

The Look AHEAD study [10,62], an intensive behavioural lifestyle intervention, evaluated the effect of BW loss on CV mortality and morbidity, in a study sample of 5154 patients with type 2 diabetes and overweight or obesity. Patients were randomized to diabetes support and education or to an intensive lifestyle intervention. The average WL was different in the two groups and the magnitude of WL was positively related to improvements in both BP and cardiovascular risk. Subjects who lost 5% to 10% of their initial BW were more likely to show a greater improvement in BP. Moreover, at 1 year, both systolic and diastolic BP declined in those patients who underwent an intensive lifestyle intervention, as compared to those who received only diabetes support and education. Only systolic BP maintained a decreased trend throughout the following progression of the study [62].

Behavioural intervention was also studied in another selected sub-population and its feasibility has been proved—in Hispanics/Latinos, for example, Rocha Goldberg et al. showed the effectiveness of educational lifestyle intervention on BW and BP control [63]; WL was observed along with a decrease in systolic BP. The ENCORE study, a large randomized controlled trial, was conducted on 144 obese or overweight hypertensive patients: subjects were randomized to a low-calorie Dietary Approach to Stop Hypertension (DASH diet) or DASH diet alone or usual diet. After four months, the subgroup assigned to DASH diet combined with a weight management program achieved both WL and a greater and significant reduction of BP [13,64,65].

### 3.2. Pharmacological Intervention

Wijkman et al. recently conducted a double-blind, placebo-controlled parallel group trial in overweight and obese patients with type 2 diabetes, randomized to receive liraglutide or placebo for 24 weeks. Compared to the placebo group, subjects who received liraglutide presented greater reduction of both BW and BP, 33% had a BP decrease of more than 5 mmHg (versus only 15% in the placebo group, (*p* < 0.01), 35% lost more than 3% in BW (vs just 3% of patients with placebo, *p* < 0.0001) and 22% of patients decreased both WL and BP versus 2% of patients in the placebo group [66]. Furthermore, the SCALE Obesity and Prediabetes trial provided evidence that overweight or obese patients, randomized to liraglutide for a period of three years, had a significant decrease in BMI, WC, systolic and diastolic BP as compared with placebo (*p* < 0.001 for all) [67]. In a multicentre, double-blind, placebo-controlled trial, Marso et al. confirmed these findings—the liraglutide group had a greater decrease in WL, as well in systolic but not in diastolic BP, than in the control group [68].

Beside the WL, other mechanisms explaining the effect of Liraglutide on BP has been hypothesized. Liraglutide treatment has been shown to increase natriuresis through a raise of natriuretic peptides [69]. Another study found increased levels of cyclic guanyl monophosphate (cGMP) and cyclic adenyl monophosphate (cAMP) which are two vasodilators and reduced plasma concentrations of angiotensinogen, renin and angiotensin after GLP-1 receptors therapy [70]. Moreover, as GLP 1 receptors are expressed in endothelial cells [71], it has been hypothesized that GLP-1 receptor agonists may improve endothelial dysfunction contributing to lower BP levels.

A review by Siebenhofer et al., of nine randomized controlled trials conducted for at least 24 weeks in hypertensive adult patients, comparing different weight reducing drugs (orlistat, sibutramine or phentermine/topiramate) to placebo, showed that treatment with orlistat is associated with WL and a significant drop in BP [72]. Sibutramine, instead, was responsible for diastolic BP increase. Phentermine/topiramate was associated to BP lowering but only one study was considered [69]. The Joint statement of the European Association for the Study of Obesity and the European Society of Hypertension confirms the positive effect of orlistat: compared to placebo, it improved both WL (more 2.7 kg) and diastolic BP, which resulted 2.2 mmHg lower [73].

### 3.3. Bariatric Surgery

In obese patients of any age, bariatric surgery has been shown to provide, together with WL, consistent improvement in systolic BP [74]. A very high number of patients treated by laparoscopic adjustable gastric banding discontinued anti-hypertensive medication or needed a lower medication dose [74,75]. Furthermore, six months after vertical sleeve gastrectomy, Seravalle et al. observed a significant reduction both in systolic BP and in sympathetic nerve conduction; interestingly, BP decline was also found to be persistent together with sympathetic inhibition 12 months after the surgical intervention [76].

As compared to lifestyle intervention, a surgical approach seems to give much more persistent and durable results [77]. In a large prospective controlled study, the Swedish Obese Subjects (SOS) study, patients were assigned to medical therapy or to different surgical procedures. A total of 4047 obese subjects were initially recruited. Surgically treated patients were matched to control subjects, who underwent a lifestyle intervention or even no treatment [78]. After a median follow-up of ten years, gastric bypass was associated with significant WL, WL maintenance and greater BP decrease, as compared both to non-surgical controls and to purely restrictive procedures such as vertical banded gastroplasty or gastric banding [79].

Nevertheless, a wide Cochrane meta-analysis by Colquitt and colleagues compared different surgical procedures, such as laparoscopic gastric bypass and laparoscopic duodenojejunal bypass with sleeve gastrectomy: however, no statistically significant differences were observed in terms of hypertension remission [80] among the different surgical procedures. In a systematic review of RCTs of bariatric surgery, Chang et al. described a 75% remission of hypertension (95% CI 62–86%) [81] independent of the type of procedure.

Considering the end organ effects of hypertension in obese patients, improvements are described after bariatric surgery. In a large review which considered CV risk factors and CV imaging in patients undergoing bariatric surgery, Vest et al. reported an echocardiographic reduction of left ventricular mass and an improvement in diastolic function, measured by E/A ratio [82]. A reduction in proteinuria, renal function decline and end stage renal disease [83] have been also observed in a recent review of observational studies by Cohen.

In a recent study by Ghanim et al., diabetic obese patients have been evaluated before and six months after Roux-en-Y gastric bypass. At the follow-up analysis, together with a significant decrease in BW and BP, a significant reduction was found in circulating vasoconstrictors (neprilysin, renin, angiotensinogen, angiotensin II and endothelin 1), whereas the vasodilator atrial natriuretic peptide (ANP) was increased [84]. Taken together, these studies show that bariatric surgery may partially explain the mechanisms of the long-term benefits of gastric bypass on BP.

## 4. Possible Mechanisms Involved in BP Reduction after Weight Loss

The explanation of the BP lowering effect of WL interventions may be identified in adipose tissue decrease. These changes may reverse the complex network of mechanisms linking obesity and hypertension (Figure 1). Visceral adipose tissue reduction, which is directly related to waist circumference reduction [61,67], may attenuate the inflammation pathway and arterial and ventricular stiffening may improve [24]. Moreover, it is well known that a visceral AT decrease, even due to a decrease of FFA release, is related to insulin resistance improvement (as also shown by the positive effect of WL, on diabetes management [10,62,66]) and a lower level of insulin may reduce systemic vasoconstriction that is partially responsible for arterial hypertension. Leptin [30] and adiponectin [34] pathways are also improved.

Since PVAT dysfunction is strongly related to obesity, it is possible to hypothesize that WL may improve PVAT functioning, by reducing the vasoconstriction effect. A reduction in vasoconstriction has been shown also after bariatric surgery [85], along with an improvement of RAAS functioning too. Weight loss may reduce renal adipose tissue as well and benefits may be found in natriuresis and sympathetic activation [45,46,49] and BP levels should consequentially lower. Unfortunately, only a few laboratory-based studies on the effect of weight loss on hypertension in the past decade have been published and new future studies are necessary to confirm possible mechanisms linking WL to the improvement of obesity-related hypertension

## 5. Conclusions

Lifestyle intervention, including weight loss, should be considered the first step in all patients with hypertension, especially if overweight and obese. All together studies aimed to show that WL induced by dietary intervention alone or associated with physical exercise or even with drugs or bariatric surgery, demonstrates a beneficial effect of WL on BP. However, the effect on BP seems to depend on the amount of WL.

Some considerations must be made. Most lifestyle intervention and pharmacological studies regarding WL and BP have been conducted on relatively small number of participants and have different follow-up lengths too. Only one study had 2 years’ follow-up [61], whereas the others had a maximum a 1-year follow-up. The only interventional study showing a long follow-up (10 years) with a persistent WL and BP reduction was the SOS study [79] but it regarded the surgical approach. Thus, studies with a longer follow-up in wider populations are needed to support these findings and to explain better the mechanisms related to the improvement of BP in obese subjects losing weight.

## Figures and Tables

**Figure 1 nutrients-11-01667-f001:**
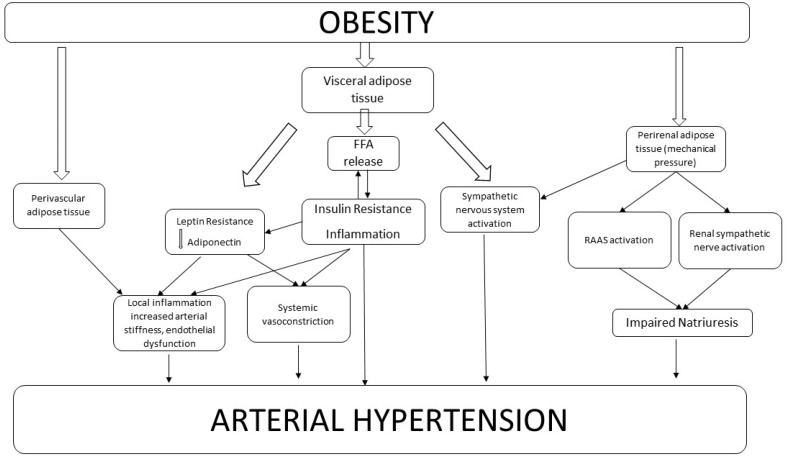
Mechanisms Linking Obesity to Hypertension.

**Table 1 nutrients-11-01667-t001:** This table summarizes selected studies from the last 10 years, showing a positive effect of weight loss (WL) on blood pressure (BP), achieved by diet and lifestyle modifications, pharmacological intervention and bariatric surgery. The table includes only studies with available data regarding number of participants, WL strategies, quantifiable mean WL and mean BP decrease, median follow-up time. * mean values referred to patients who achieved the major waist circumference reduction.

Author, Year	Number of Participants	WL Intervention	Mean WL ΔBW (Kg) ΔBMI (kg/m^2^)		Mean BP Reduction (mmHg)	Median Follow up
**Diet and Lifestyle modification**						
**Blumenthal, 2010** **(*ENCORE study*)**	144	DASH diet alone	−0.3		11.2 (SBP) 7.5 (DBP)	4 months
		DASH diet plus weight management	−8.7		16.1 (SBP) 9.9 (DBP)	
**Rocha-Goldberg, 2010**	17	behavioral intervention	1.5 ± 3.2 lb		10.4 ± 10.6 (SBP)	6 weeks
**Rothberg, 2017**	344	behavioral intervention		−6 ± 3	8 (SBP) *	6 months
	170	behavioral intervention		−5 ± 4	8 (SBP) *	2 years
**Straznicky, 2011**	59	dietary and moderate-intensity aerobic exercise	−7.1 ± 0.6 (dietary)	−2.4 ± 0.2 (dietary)	10±2 (SBP)	12 weeks
			−8.4 ± 1.0 (dietary + exercise)	−2.8± 0.3 (dietary + exercise)		
**Wing, 2011** ***(look AHEAD study)***	5154	intensive lifestyle intervention or diabetes support and education	−4.8 ± 7.6		2.40 (DBP) 4.76 (SBP)	1 year
**Pharmacological intervention**						
**Marso, 2016**	9340	Pharmacologic (liraglutide vs placebo)	2.3 kg *higher in Liraglutide group*		1.2 (SBP) *lower in Liraglutide group*	36 weeks
**Wijkman, 2019**	124	Pharmacologic (liraglutide vs placebo)	>3%		9.2 (SBP)	24 weeks
**Bariatric Surgery**						
**Ghanim, 2018**	15	Surgery (RYGB)		−11.7	11 (SBP)	6 months
**Hallersund, 2013** **(*SOS study*)**	2473 (277 gastric bypass, 1064 purely restricted proedures, 1132 control)	Surgery (GBP, VBG/B)		−10.1 (GBP group)	−5.1 (SBP) −5.6 (DBP) (GBP group)	10 years
**Seravalle, 2014**	20 (10 surgery + 10 control)	Surgery (vertical sleeve gastrectomy)		−9.1 ± 1.4	10.2 ± 4.5 (SBP)	6 months
				−10.8 ± 1.6	13.9 ± 5.0 (SBP)	1 year

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
