# Peer review of "Weight Loss and Hypertension in Obese Subjects"

_nutrients, 2019, doi:10.3390/nu11071667_

Reviewer 1 Report

Suggestions for edits are as follows:

Lines 28-29   ….increasing with BMI…..

Line 30 --  These percentages are……

Line 31--…in the general population:

Line 33-- …and it is associated with higher risk…..

Line 40—Regardless of the grade….

Line 43---….a moderate WL has a…..

Line 71---….independent of other the other classical risk factors.

Line 103---…often associated with…..

Line 128----… in obese subjects…

Line 133---…..kidneys seem to have “lipotoxic”….

Line 167----…..heart rate…..

Line 228----The ENCORE study……..was conducted on 144 obese…..

Line 234---…..number of participants….

Line 242----About 33% had a BP decrease more than……

Line 254--- A review by Siebenhofer et al of nine……

Line 264--- In obese patients of any age,…

Line 307----Some considerations must be made……

Author Response

We thanks reviewer 1 for the suggestions and the comments.

We replied point by point as follows:

Lines 28-29   ….increasing with BMI…..

Line 30 --  These percentages are……

Line 31--…in the general population:

Line 33-- …and it is associated with higher risk…..

Line 40—Regardless of the grade….

Line 43---….a moderate WL has a…..

Line 71---….independent of other the other classical risk factors.

Line 103---…often associated with…..

Line 128----… in obese subjects…

Line 133---…..kidneys seem to have “lipotoxic”….

Line 167----…..heart rate…..

Line 228----The ENCORE study……..was conducted on 144 obese…..

Line 234---…..number of participants….

Line 242----About 33% had a BP decrease more than……

Line 254--- A review by Siebenhofer et al of nine……

Line 264--- In obese patients of any age,…

Line 307----Some considerations must be made……

All these editing changes have been done and the paper has been widely reviewed and edited by a native professional English speaker.

Reviewer 2 Report

This review discusses some very relevant topics regarding the mechanisms linking obesity and hypertension and the effect of weight loss on hypertension. Some sections of the review are very comprehensive and cover much of the new and exciting literature available in these areas, in particular the last section that is on the effect of weight loss on hypertension in obesity.  However, some of the sections do seem quiet disjointed from one another, in particular it is unclear how the section on mechanisms of hypertension in obesity relates to the section on weight loss. 

Major comments:

1.     The language and grammar needs extensive editing (I suggest having it edited by a professional). It is difficult to clearly understand what the authors are trying to say at times due to the poor grammar and structure of sentences. The review also need to flow better between sentences and paragraphs, it seems a little disjointed at times (and there are also many paragraphs throughout, which consist of just one sentence, paragraphs should generally have at least 3 sentences). 

2.     I suggest the removal of section 2. “Hemodynamics and structural changes in obese subjects with hypertension”. This paragraph is very shot, and is essentially just a summary of a previous (and very comprehensive) review on this topic. The authors could keep this reference and state that they will not cover this in detail, and reference (several) of the available reviews on this topic, such as the one they have referenced here. 

3.     There are many studies that examine the immune response in PVAT in hypertension. I suggest the authors reference some of these published studies, rather than an abstract from a recent meeting. This section could also be expanded to include more detail regarding the immune response in PVAT during hypertension. 

4.     There does seem to be a lack of laboratory-based studies on the effect of weight-loss on hypertension in the past decade (most are on the effect of weight-gain on hypertension). Maybe this is something the authors could comment on?

5.     Tables 1-3 could be amalgamated into a single table. This would help readers compare the 3 different types of weight loss too. 

6.     The first half the review focuses quiet heavily on mechanism, however section 4, barely mentions mechanism. It would be good if the authors could either discuss some of the mechanisms at play in the various weight loss techniques. This would help link the second half of the review, with the first half.

Minor comments:

1.     Abbreviations: PVAT does not need to be abbreviated in the abstract, it is only used once. Abbreviations in main text should be defined upon first use in main text. i.e. BP

2.     This statement needs a reference: Moreover, the ectopic fat accumulation in and around the kidneys seems to play “lipotoxic” effects on the kidneys through increased oxidative stress, mitochondrial dysfunction, and endoplasmic reticulum stress.

3.     Can the authors please find the original reference for the following statement: “Although insulin is a vasodilator hormone, insulin resistance, would reduce insulin vasodilation capacity, reducing the nitric oxide (NO) production by endothelial cells”

4.     There are some typos throughout the manuscript, such as “aldosterone” line 152

Author Response

Response to Reviewer 2 Comments

Major comments:

1.     The language and grammar need extensive editing (I suggest having it edited by a professional). It is difficult to clearly understand what the authors are trying to say at times due to the poor grammar and structure of sentences. The review also needs to flow better between sentences and paragraphs, it seems a little disjointed at times (and there are also many paragraphs throughout, which consist of just one sentence, paragraphs should generally have at least 3 sentences).

As requested, the paper has been widely reviewed and edited by a native professional English speaker (Prof Mark Newmann).

2.     I suggest the removal of section 2. “Hemodynamics and structural changes in obese subjects with hypertension”. This paragraph is very short, and is essentially just a summary of a previous (and very comprehensive) review on this topic. The authors could keep this reference and state that they will not cover this in detail, and reference (several) of the available reviews on this topic, such as the one they have referenced here. 

The section has been removed and, as suggested, the reference has been left explaining that we have not covered this section in detail, as suggested (page 2 line 62-67).

3.     There are many studies that examine the immune response in PVAT in hypertension. I suggest the authors reference some of these published studies, rather than an abstract from a recent meeting. This section could also be expanded to include more detail regarding the immune response in .

We agree with reviewer's suggestion. A new paragraph with new references has been added to this section in the new version of the paper (Page 3, line 122-129).

4.     There does seem to be a lack of laboratory-based studies on the effect of weight-loss on hypertension in the past decade (most are on the effect of weight-gain on hypertension). Maybe this is something the authors could comment on?

In the new section explaining the possible mechanism relating obesity and hypertension we added a comment about this point (Page 10 line 301).

 5.     Tables 1-3 could be amalgamated into a single table. This would help readers compare the 3 different types of weight loss too

Following the reviewer’s suggestion, we have amalgamated the 3 tables in 1 now.

6.     The first half the review focuses quiet heavily on mechanism, however section 4, barely mentions mechanism. It would be good if the authors could either discuss some of the mechanisms at play in the various weight loss techniques. This would help link the second half of the review, with the first half. 

 We have added a paragraph about the possible mechanisms involved in the BP reduction after WL (Page 10 line 301).

Minor comments:

1.     Abbreviations: PVAT does not need to be abbreviated in the abstract, it is only used once. Abbreviations in main text should be defined upon first use in main text. i.e. BP

We have changed the text accordingly

 2.     This statement needs a reference: Moreover, the ectopic fat accumulation in and around the kidneys seems to play “lipotoxic” effects on the kidneys through increased oxidative stress, mitochondrial dysfunction, and endoplasmic reticulum stress.

We have added a reference for this sentence (page 4 line 138, reference 48)

 3.     Can the authors please find the original reference for the following statement: “Although insulin is a vasodilator hormone, insulin resistance, would reduce insulin vasodilation capacity, reducing the nitric oxide (NO) production by endothelial cells”

We added references 18 and 19 (page 2 line 77)

 4.     There are some typos throughout the manuscript, such as “aldosterone” line 152

Typos have been revised and the manuscript has been edited by a native professional English speaker.

Round  2

Reviewer 2 Report

1. There are still many paragraphs throughout, which consist of just one sentence. As mentioned previously, paragraphs should generally have at least 3 sentences. if the authors could please ensure the stand alone sentences are joined to nearby paragraphs.

2. Section 1, 2nd paragraph, what do the authors mean by office measurement?

3. Could the authors please define grade 2 and 3 hypertension

4. The statement "In obese patients a metabolic syndrome, a moderate WL improves renal function [11] and may lead to a 15% reduction of all-cause mortality." is grammatically incorrect.

5. It is unclear how the statement "Nevertheless, in obese animal models, administration of clonidine or alpha or beta blockers have been shown to prevent an increase in BP" relates to section 2.5

6. The pharmacological agents section could be expanded. Why are there different effects between these drugs? What are the modes of action?

Author Response

Response to Reviewer 2 Comments

1. There are still many paragraphs throughout, which consist of just one sentence. As mentioned previously, paragraphs should generally have at least 3 sentences. if the authors could please ensure the stand alone sentences are joined to nearby paragraphs.

Following the reviewer’s suggestion now the paragraphs have at least 3 sentences each.

2. Section 1, 2nd paragraph, what do the authors mean by office measurement?

As explained in the new version of the paper, office measurements are the measurements obtained during medical evaluation.

3. Could the authors please define grade 2 and 3 hypertension

Grade 2 and 3 hypertension are now defined in the new version.

4. The statement "In obese patients a metabolic syndrome, a moderate WL improves renal function [11] and may lead to a 15% reduction of all-cause mortality." is grammatically incorrect.

The sentence has been corrected in the new version of the paper.

5. It is unclear how the statement "Nevertheless, in obese animal models, administration of clonidine or alpha or beta blockers have been shown to prevent an increase in BP" relates to section 2.5

Following the reviewer’s suggestion, we decide to remove this sentence in the new version of the paper.

6. The pharmacological agents section could be expanded. Why are there different effects between these drugs? What are the modes of action?

Following reviewer’s suggestion, a new paragraph explaining the mechanism related to the effect of lariglutide on BP reduction  (besides WL) has been added. In the new version of the paper 3 new references (70,71, 72) have been added related to this new paragraph.